# Effect of Intermittent Hypoxic Training on Selected Biochemical Indicators, Blood Rheological Properties, and Metabolic Activity of Erythrocytes in Rowers

**DOI:** 10.3390/biology11101513

**Published:** 2022-10-15

**Authors:** Aneta Teległów, Mateusz Mardyła, Michał Myszka, Tomasz Pałka, Marcin Maciejczyk, Przemysław Bujas, Dariusz Mucha, Bartłomiej Ptaszek, Jakub Marchewka

**Affiliations:** 1Department of Health Promotion, Institute of Basic Sciences, University of Physical Education in Krakow, 31-571 Krakow, Poland; 2Institute of Biomedical Sciences, University of Physical Education in Krakow, 31-571 Krakow, Poland; 3AZS AWF Krakow Sports Club, 31-571 Krakow, Poland; 4Department of Sports Theory and Anthropomotorics, University of Physical Education in Krakow, 31-571 Krakow, Poland; 5Institute of Applied Sciences, University of Physical Education in Krakow, 31-571 Krakow, Poland; 6Department of Rehabilitation in Traumatology, Institute of Clinical Rehabilitation, University of Physical Education in Krakow, 31-571 Krakow, Poland

**Keywords:** rowers, training, hypoxia, biochemical indicators, blood rheology, erythrocyte metabolic activity, maximal oxygen uptake

## Abstract

**Simple Summary:**

Rowing is among the oldest sports disciplines. Involving multiple muscle groups, it poses a considerable challenge to athletes and amateurs. To improve training efficiency, the hypoxic method (decreasing arterial oxygen concentration) is sometimes applied. The study determined the effect of 3-week hypoxic training on blood indicators in competitive rowers. The reported findings demonstrated that the phenomenon of hypoxia could be an extremely effective training measure in rowers as it exerted a beneficial impact on the athletes’ performance.

**Abstract:**

The study assessed the effect of 3-week intermittent hypoxic training on blood biochemical indicators (blood morphology, fibrinogen), blood rheological properties (erythrocyte deformability, aggregation), erythrocyte enzymatic activity (acetylcholinesterase), and maximal oxygen uptake in competitive rowers. Fourteen male rowers were divided into two equal groups: experimental, training on ergometers under normobaric hypoxia (FiO_2_ = 16.0%), and control, training on ergometers under normoxia (FiO_2_ = 21%). Fasting blood was taken before and after training. A significant between-group difference in neutrophil levels before training was noted and a significant decrease in white blood cells in the hypoxia group. Both groups exhibited an increase in elongation index. In the normoxia group, a significant increase in erythrocyte aggregation amplitude was revealed. No significant changes occurred in the other biochemical indicators or those evaluating erythrocyte metabolic activity. Normobaric hypoxia increased erythrocyte deformability, improving blood rheological properties. Maximal oxygen uptake significantly increased only in the experimental group.

## 1. Introduction

Rowing is among the oldest sports disciplines. It is an endurance and strength discipline, with a starting effort duration ranging from 5.5 to almost 8 min, depending on the boat type [1]. Energy supply during starting efforts is largely based on aerobic metabolism, which accounts for approximately 87% of the total energy expenditure [2]. Elite rowers represent high levels of maximal aerobic power and/or lactate tolerance during a race [3]. During the training, it is important to continuously control the training load and physiological and biochemical blood parameters [4]. A relatively large part of the preparatory period is devoted to training with the use of rowing ergometers [5]. Monitoring tests are very important to control fatigue in order to avoid overtraining [6]. In monitoring tests, it is most crucial to perform blood assays to determine exercise intensity, skeletal muscle damage, indicators of overtraining, body hydration, blood rheology indicators. Major determinants of blood rheology under physical effort are haematocrit and fibrinogen concentrations and plasma viscosity; their increases affect blood haemoconcentration [7]. Exercise training induces adaptations in erythrocyte aggregation and raises erythrocyte deformability by increasing the circulating erythropoietin. Repeated exercise during consecutive days leads to a chronic ‘autohaemodilution’, which results in a low baseline haematocrit–low baseline viscosity patterns. In contrast, overtraining syndrome is associated with a reversal of the ‘autohaemodilution’ phenomenon, with haematocrit, plasma, and blood viscosity increasing above the levels observed in the trained state [8]. Hypoxia also enhances erythrocyte aggregation and plasma viscosity. In addition, exposure to hypoxia can increase haematocrit levels and modulate erythrocyte deformability. Hypoxia dose, however, is crucial: beneficial effects of intermittent hypoxia at a moderate dose compared with harmful effects of chronic continuous or intermittent hypoxia and of hypoxia applied at excessive doses were noted [9].

Implementing hypoxia in the training plan during the preparatory period is widespread as a legitimate supportive tool to improve training efficiency [10]. Numerous ‘live low–train high’ training methods have been described that involve the application of systemic and local hypoxia stimuli, or a combination of both, for performance enhancement in many disciplines [11]. A modification of these methods is the concept of intermittent hypoxic training (IHT). It is this training model that allows one to avoid the adverse effects of staying at natural heights, such as altitude sickness, pulmonary oedema, or cardiac arrhythmias [12]. Training under hypoxia conditions triggers a number of adaptive mechanisms [13], with hypoxia-inducible factor playing a critical role in adaptation to hypoxia. Under its influence, the rate of erythropoiesis in the bone marrow is accelerated [14]. Although many studies have demonstrated no effect on increasing physical capacity in competitive athletes after IHT [15,16], there are also data indicating improvements in aerobic [17,18] and anaerobic [19,20] capacity. The literature also provides contradictory data on the change in haematological indicators under the influence of IHT. However, a study published in 2010 [20] shows an improvement in basic haematological indicators, i.e., an increase in haemoglobin and haematocrit. Additionally, numerous publications reflect an impact of different forms of hypoxia on physical capacity and other indicators among competitive rowers [21].

The aim of the presented study was to assess the effect of a 3-week IHT on blood biochemical indicators (blood morphology, fibrinogen), blood rheological properties, (erythrocyte deformability and aggregation), erythrocyte enzymatic activity (acetylcholinesterase), and maximal oxygen uptake (VO_2_max) in advanced and elite competitive rowers. The available literature related to the above-mentioned objective is scarce. In our study, we wanted to provide a broader perspective of the analyses with respect to numerous blood rheological and biochemical indicators and enzyme activities. Rowing poses a considerable challenge to athletes and amateurs alike. It involves multiple muscle groups, which increases capacity and positively influences heart function. We hypothesized that IHT would improve blood rheological properties and erythrocyte enzymatic activity in rowers, which could then translate into enhanced exercise capacity.

## 2. Material and Methods

### 2.1. Study Groups

The study involved 14 male advanced and elite competitive rowers, including 3 from the Polish national team. Among the subjects, 1 was a participant in the Senior World Championships and a medallist at the Polish Championships, 2 were medallists at the Polish Championships, and the remaining 11 were finalists at the Polish Championships in various competitions. The sample size was not calculated. All participants in the study (control and IHT group) had to perform the same physical training, and the only difference was the inclusion of IHT in one group. For this reason, the athletes were from one sports club, and the study included those who gave their written consent. The inclusion criteria were training experience of at least 3 years, age of 18–30 years, and regular practice of rowing. The exclusion criteria were as follows: tobacco smoking, history of musculoskeletal injuries within the previous 3 months, and training breaks longer than 1 month (fewer than 4 training units per week). The investigated athletes had certificates of appropriate medical examinations allowing them to practise sport. They did not undergo continuous specialist treatment, and the diet of each player was not modified during the study. All the competitors resided in the central part of Lesser Poland, while their most recent stay in the mountains was 3 months earlier (in December). During the 3 weeks of the experiment, the training loads did not change.

The participants were randomly divided Into 2 groups:The experimental group (*n* = 7) of male competitive rowers training on ergometers under normobaric hypoxia conditions (IHT) (FiO_2_ = 16.0%), corresponding to an altitude of approximately 2400 m above sea level. The hypoxic conditions were achieved with the use of Everest Summit II hypoxicators. During the training, the athletes breathed through special masks that were connected to generators by air tubes.The control group (CON) (*n* = 7) of male competitive rowers training on ergometers under normoxia conditions (FiO_2_ = 21%).

The analyses of somatic indicators were performed with a Tanita BC-601 analyser (Tanita Inc., Tokyo, Japan) (Table 1).

### 2.2. Rowing Performance and Intermittent Hypoxic Training

Before and after the training program, a standard rowing test of 2000 m in normoxia was performed in both groups on the Concept II indoor rower (model D, VT, Morrisville, VT, USA). This allowed us to determine the current capacity levels of the examined athletes, as well as to indirectly estimate VO_2_max, in accordance with the following formula [22]:VO_2_max [L/min] = 1.682 + 0.0097 × MP(1)
where MP denotes the mean power (in watts) achieved in the standard 2 km test.

Training under hypoxia conditions was carried out during February and March, in the pre-season preparation period, with the use of Everest Summit II hypoxicators (Hypoxico Inc., New York, NY, USA), in the Central Research and Development Laboratory, University of Physical Education in Krakow, Poland. The room had a constant temperature of 20 °C and humidity of approximately 45%. In 3 week-long microcycles, the participants performed simulated training at an altitude of approximately 2400 m. On each training day, the percentage concentration of oxygen in the breathing air was monitored with a special Handigen Oxygen Monitor device (Hypoxico Europe, Bickenbach, Germany). The training program was identical in both study groups (IHT and normoxia). Training on ergometers was held on Mondays, Wednesdays, and Fridays. A single exposure to hypoxia lasted for 60–70 min, depending on the training day. The breathing mask was put on before the warm-up and taken off on completion of the main part of the training, so that during the calm-down part the subjects breathed with atmospheric air. The authors did not directly examine the fraction of air taken in. However, the oxygen content in the air flowing into the athlete’s sealed mask was assessed; in addition, the rowers’ saturation was determined with a fingertip pulse oximeter every 10 min.

The time outside the ergometer training sessions (under hypoxia and normoxia) was spent on specialized training on the water (2 times per week), strength training (2 times per week), general development training (1 time per week), cross-country running (1 time per week), and team games (2 h) (Table 2).

Prior to training, the lactate threshold was determined in a graded test. In both groups, the test started at an intensity of 140 W; an increase by 30 W was applied every 3 min until refusal [23]. A rise in lactate concentration of more than 0.5 mmol/L between successive loads was considered the point of lactate threshold occurrence. The power volume at the lactate threshold obtained by this method corresponds with the power volume at the change point in oxygen uptake as described by Zoladz et al. [24], which may reflect a change in effort energy and be a good indicator of the effectiveness of the applied training. The measurements were taken in capillary blood by using a Lactate Scout+ device (SensLab, Leipzig, Germany). During the test, heart rate was also recorded with a Garmin Fenix 3 sports watch (Garmin Ltd., Schaffhausen, Switzerland) to determine heart rate at the lactate threshold.

A single unit of training on the ergometer lasted for 50–60 min and involved a warm-up (5 min), the main part depending on the planned training (40–50 min), and the calming part (5 min). On Mondays and Fridays, the athletes performed a 50 min continuous training (workload of 90% of the lactate threshold power for the IHT group and 100% of the lactate threshold power for the normoxia group). The mean training volume measured by the distance rowed on these days was 12.28 km in the IHT group and 13.02 km in the normoxia group. On Wednesdays, the participants performed interval training 5 × 12 min with 4 min breaks. The 12 min sections were covered with varying intensity: 4 min at 80/90% of the lactate threshold power (in the IHT group and in the normoxia group, respectively), 4 min at 100/110% of the lactate threshold power, and 4 min at 80/90% of the lactate threshold power. The mean distance covered during the ergometer training was 15.54 km and 16.68 km in the IHT group and in the normoxia group, respectively.

During training, blood saturation was measured in the examined rowers by using a pulse oximeter (Contec CMS50D, Contec Medical Corporation, Qinhuangdao, China). In each training session, it equalled 78–86% in the IHT group and 88–97% in the normoxia group.

### 2.3. Blood Sample Collection

In both study groups, fasting blood was taken before the 3-week training commencement, and then after completion of the 3-week training, under normobaric hypoxia (IHT) (*n* = 7) and under normoxia (*n* = 7). The blood samples were collected from an ulnar vein, in the total amount of approximately 15 mL, by a qualified nurse at the Blood Physiology Laboratory of the University of Physical Education in Krakow. The material analyses were performed in the Diagnostyka S.A. laboratory in Krakow.

### 2.4. Morphological, Rheological, and Biochemical Assessments

Complete blood count was performed with an ADVIA 2120i analyser (Siemens Healthineers, Erlangen, Germany) and involved white blood cell count [×10^9^/L], neutrocyte count [×10^9^/L], lymphocyte count [×10^9^/L], monocyte count [×10^9^/L], eosinocyte count [×10^9^/L], basophil count [×10^9^/L], red blood cell count [×10^12^/L], haemoglobin concentration [g/dL], haematocrit [%], mean corpuscular volume [fL], mean corpuscular haemoglobin [pg], mean corpuscular haemoglobin concentration [g/dL], red blood cell distribution width [fL], platelet count [×10^9^/L], mean platelet volume [fL], procalcitonin concentration [%], and platelet distribution width [fL]. As for blood rheology, the aggregation (aggregation index [%], amplitude and total extent of aggregation [arbitrary units], and half-time of total aggregation [s]) and deformability of red blood cells (EI, elongation index) were tested with the Lorrca Maxsis osmoscan (Lorrca^®^, RR Mechatronics, AN Zwaag, The Netherlands) and the method by Hardeman [25,26]. The mean EI was plotted versus the corresponding shear stress of 0.30–60.00 Pa. The coagulation parameter of fibrinogen [g/L] was determined with a BCS Siemens coagulation analyser. The metabolic activity of erythrocytes was described by means of the activity of the enzyme acetylcholinesterase (EC 3.1.1.7). Acetylcholinesterase activity and reduced glutathione level were determined with the spectrophotometric method as described by Beutler [27].

### 2.5. Statistical Analysis

The research results are presented as means and standard deviations. Data distribution was checked by using the Shapiro–Wilk test. All variables exhibited normal distribution. In order to assess the significance of differences between groups before and after the training intervention, two-way analysis of variance (ANOVA) with repeated measures was applied; it analysed the main factors: group (IHT versus CON) and time (pre versus post), as well as the interaction between these factors. Then, a post hoc analysis was carried out with Tukey’s test. Homogeneity of variance within the groups was tested via Levene’s test (the variance of the analysed parameters was similar in both groups). Statistical analysis of the results was performed with the Statistica 12.0 software (StatSoft, Tulsa, OK, USA). The differences in all analysed indicators were considered statistically significant at the level of *p* < 0.05.

## 3. Results

No significant differences in power levels between the normoxia and IHT conditions were demonstrated (*f* = 1.15, *p* = 0.30), whereas the change between the tests (pre-post) turned out statistically significant (*f* = 8.80, *p* = 0.01). The post hoc analysis revealed a significant difference in power between the tests only in the IHT group (*p* = 0.03); in the normoxia group, the difference was insignificant (*p* = 0.77). No interaction was found between the factors (*f* = 2.60, *p* = 0.13).

With regard to VO_2_max, no significant differences were observed between the normoxia and IHT conditions (*f* = 1.10, *p* = 0.32), whereas the change between the tests (pre-post) turned out statistically significant (*f* = 7.21, *p* = 0.02). The post hoc analysis revealed a significant difference in VO_2_max between the tests only in the IHT group (*p* = 0.049); in the normoxia group, the difference was insignificant (*p* = 0.82). No interaction was found between the factors (*f* = 2.19, *p* = 0.16) (Figure 1).

The investigated groups did not differ in the level of the examined blood morphology indicators. There were significant changes over time (between the first and the second test) in the levels of white blood cells (*f* = 6.84, *p* = 0.02) and lymphocytes (*f* = 11.25, *p* = 0.007), while no statistically significant changes over time were observed for the other parameters. A significant interaction (group × time) was found only for neutrophils (*f* = 5.18, *p* = 0.04) and eosinophils (*f* = 6.78, *p* = 0.02). The post hoc analysis revealed significant between-group differences in neutrophil levels (*p* = 0.05) before training and a significant decrease in white blood cells in the group of rowers training in hypoxia (*p* = 0.05). For the other studied variables, no significant between-group differences were recorded either before or after training, and there were no significant changes over time (Table 3).

The investigated groups did not differ in the levels of the examined red blood cell EI. There were significant changes over time (between the first and the second test) in the level of EI at a shear stress of 4.24 (*f* = 5.45, *p* = 0.04), at a shear stress of 8.23 Pa (*f* = 31.73, *p* = 0.0002), at a shear stress of 15.95 Pa (*f* = 65.55, *p* = 0.00001), at a shear stress of 30.94 Pa (*f* = 131.4, *p* = 0.00), and at a shear stress of 60.00 Pa (*f* = 197.4, *p* = 0.00), while no statistically significant changes over time were observed for the other parameters. A significant interaction (group × time) was found only for EI at a shear stress of 30.94 Pa (*f* = 5.4, *p* = 0.04). The post hoc analysis revealed a statistically significant increase in EI at a shear stress of 8.23 Pa (*p* = 0.003), an increase in EI at a shear stress of 15.95 Pa (*p* = 0.0005), an increase in EI at a shear stress of 30.94 Pa (*p* = 0.0002), and an increase in EI at a shear stress of 60.00 Pa (*p* = 0.0002) in the group of rowers training in hypoxia. In the normoxia group, the post hoc analysis indicated a statistically significant increase in EI at a shear stress of 8.23 Pa (*p* = 0.04), an increase in EI at a shear stress of 15.95 Pa (*p* = 0.002), an increase in EI at a shear stress of 30.94 Pa (*p* = 0.0005), and an increase in EI at a shear stress of 60.00 Pa (*p* = 0.0002). For the other studied variables, no significant between-group differences were recorded either before or after training, and there were no significant changes over time (Table 4 and Figure 2).

The investigated groups did not differ in the level of the examined blood aggregation indicators. There were significant changes over time (between the first and the second test) in the level of erythrocyte aggregation amplitude (AMP) (*f* = 32.16, *p* = 0.0002), while no statistically significant changes over time were observed for the other parameters. A significant interaction (group × time) was found only for AMP (*f* = 4.82, *p* = 0.05). The post hoc analysis revealed a statistically significant increase in AMP (*p* = 0.001) in the group of rowers training in normoxia (the control group). For the other studied variables, no significant between-group differences were recorded either before or after training, and there were no significant changes over time (Table 5 and Figure 3, Figure 4 and Figure 5).

The investigated groups did not differ in fibrinogen concentrations. No significant between-group differences were recorded either before or after training, and there were no significant changes over time or interaction (group × time). The post hoc analysis did not reveal significant differences (Table 6).

The investigated groups did not differ in acetylcholinesterase activity. There were significant changes over time (between the first and the second test) in the levels of acetylcholinesterase (*f* = 6.98, *p* = 0.024) (Table 7).

## 4. Discussion

Our study involved competitive athletes—rowers. The study aim was to assess the impact of a 3-week IHT on blood biochemical indicators (blood morphology, fibrinogen), blood rheological properties (erythrocyte deformability and aggregation), and erythrocyte enzymatic activity (acetylcholinesterase) in rowers. The study focused on the practical application of hypoxia training among rowers against the background of changes in selected blood biochemical and rheological indicators. Appropriately selected training increases exercise capacity by enhancing the results obtained in a discipline-specific stress test, as well as improves blood rheological properties without accompanying unfavourable changes in biochemical indicators. Even though VO_2_max was not measured directly, the test results achieved after the 3-week training provide an indirect indication of an increase in VO_2_max. The phenomenon of hypoxia is widely applied in sport to enhance athletes’ exercise capacity. No reports were found in the literature on the effect of training in normobaric hypoxia on the rheological, morphological, or biochemical properties of blood among competitive rowers.

The main research hypothesis was that training under hypoxia would induce beneficial changes in the blood rheological profile and improve capacity in competitive rowers. This impact may be maintained for up to several weeks after hypoxia training. The blood cell deformability, which expanded in both investigated groups, is indicative of correct training planning in the period before the immediate start preparation. The performed graded tests demonstrate an increase in the effort capacity and VO_2_max, which may be an indirect consequence of an improvement in red blood cell elongation and thus of a raised efficiency of oxygen delivery to working tissues.

The phenomenon of red blood cell deformability plays an important role in blood cell flow through the vessels; it indicates an increase in tissue oxygenation and an extended lifetime of erythrocytes [8]. Hypoxia exposure can increase haematocrit levels and modulate erythrocyte deformability in a dose-dependent manner, i.e., beneficial effects of intermittent hypoxia at moderate doses versus deleterious effects of chronic continuous or intermittent hypoxia or hypoxia administered at excessive doses were observed [9]. According to Baskurt et al. [28], in well-trained athletes, increased blood fluidity may enhance oxygen delivery to muscles during exercise. After 3-week training, both groups of rowers (training in normobaric hypoxia and in normoxia) exhibited improved red blood cell deformability in the blood vessel system, but AMP was raised only among those training in normoxia. This potentially results from differences in body hydration and dehydration, which considerably influence blood rheological properties. The formation of blood cell aggregates is favoured by an increase in fibrinogen. Erythrocyte aggregation is induced by macromolecular bridges, which include fibrinogen. Our study did not reveal statistically significant changes regarding fibrinogen. In contrast, in a study by Moon et al. [29], submaximal effort on an ergometer undertaken in hypoxia did not induce changes in erythrocyte aggregation indicators. As implied by Monnier et al. [30], hydration and dehydration are determinants of both blood rheology and exercise efficiency.

Staying and training in high-altitude or hypoxia conditions increase the plasma volume, which contributes to the reduction in plasma viscosity, thus lowering the blood flow resistance in the vessels. Cheng et al. [31] found that residents of high-altitude areas exhibited higher haematocrit levels and greater deformability of red blood cells. Data on the effect of IHT on resting red blood cell deformability properties are lacking. In a 2016 study, reduced erythrocyte deformability was observed during effort under conditions with decreased oxygen concentration (16.5%, 14.5%, 12.8%, 11.2%), as well as in the short restitution period [29]. The lowest EI value was reported in the pre-exercise period, when the oxygen concentration in the hypoxic chamber equalled 11.5%. Similar results were achieved by Grau et al. [32], although they subjected the participants only to acute exposure to various hypoxic conditions. One of the crucial factors indirectly determining deformability is nitric oxide; however, deformability may also be affected by mean corpuscular volume. As recently indicated by von Tempelhoff et al. [33], a decrease in mean corpuscular volume and an increase in mean corpuscular haemoglobin concentration contribute to lower deformability and restrained passage through the microcirculation vessels. Many authors emphasize the individual variability of changes in haematological indicators in response to hypoxia [34]. Wehrlin and Marti [35] observed that athletes who lived at 2500 m above sea level and trained at lower altitudes for 24 days exhibited increased haemoglobin mass and red blood cell volume as compared with those who stayed and trained at low altitudes.

In the present study, no statistically significant changes were found for the indicators of the red cell and platelet systems. In the white cell system, the post hoc analysis revealed significant between-group differences in neutrophil levels (*p* = 0.05) before training and a significant decrease in white blood cells among rowers training in the hypoxia conditions (*p* = 0.05). This may result from leukocyte apoptosis induced by physical effort in hypoxia and the consequent shorter leukocyte lifetime.

One should emphasize that a high content of polyunsaturated fatty acids in red blood cells, the fact of transporting oxygen bound with haemoglobin, and the lack of repair systems resulting from the absence of cell nucleus and protein synthesis organelles make red blood cells particularly vulnerable to reactive oxygen species. In addition, they are exposed to mechanical and osmotic stress accompanying about 150,000 cycles in the circulation [36,37]. Owing to the presence of many enzymes and proteins in the cell membrane, erythrocytes can effectively counteract the undesirable consequences of oxidative stress that may occur as a result of an overly intense training process. In the presented study, either in hypoxia or in normoxia conditions, no statistically significant changes were observed in the examined rowers for acetylcholinesterase. One of the more important defence mechanisms appears to be related to an enzyme occurring on the surface of the erythrocyte membrane, acetylcholinesterase, whose activity changes may correspond to the degree of oxidative stress exposure [38].

### Limitation of the Study

The study is limited by its relatively small sample size. However, its value consists of the fact that it involved advanced and elite rowers, including national rowing team members. As there are few rowers at this competitive level, performing similar research with a larger sample would be substantially problematic. Enrolling competitive athletes in further research should also include other sports disciplines, with a different nature of effort. Investigating changes in blood and plasma viscosity, changes in iron metabolism, and counts of reticulocytes, which are produced in increased numbers under hypoxia, could also contribute to obtaining a complete picture of changes that may affect blood flow and effective oxygen delivery to tissues. In addition, blood tests performed in different periods of the annual training plan among athletes could help understand the effect of training and competition themselves on modifications in rheological indicators and erythrocyte properties.

## 5. Conclusions

During the 3 weeks of training in hypoxia, a significant improvement was observed in aerobic capacity indirectly measured with a discipline-specific test, whereas no such effect was reported in the normoxia group. Since there were no differences in red blood cell indicators, other, more difficult-to-measure factors may be involved, such as increased mitochondrial density, improved muscle buffer capacity, or enhanced lung ventilation associated with this type of training.

## Figures and Tables

**Figure 1 biology-11-01513-f001:**
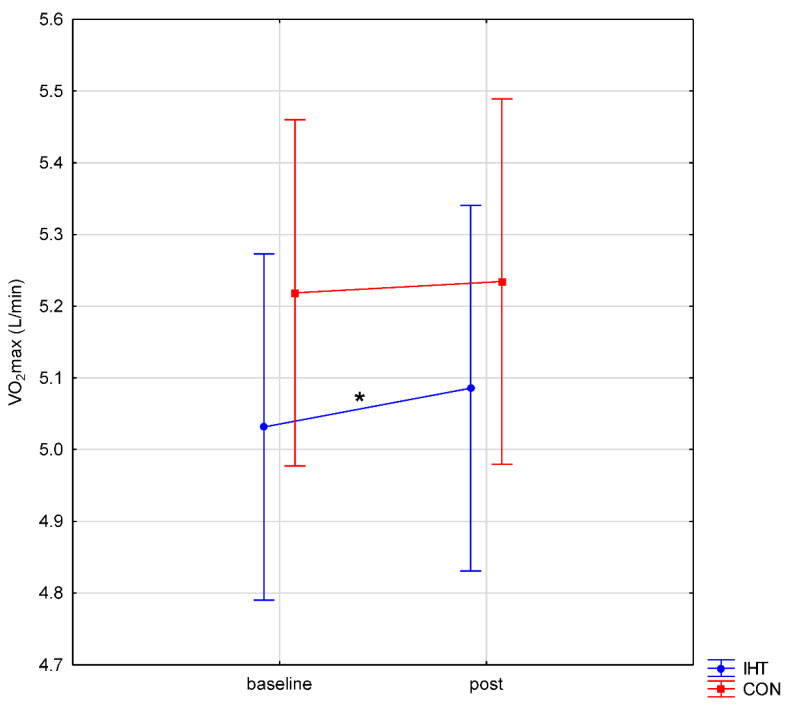
Maximal oxygen uptake (VO_2_max) during intermittent hypoxic training (IHT) and normoxia (CON, control) training (*: *p* < 0.05).

**Figure 2 biology-11-01513-f002:**
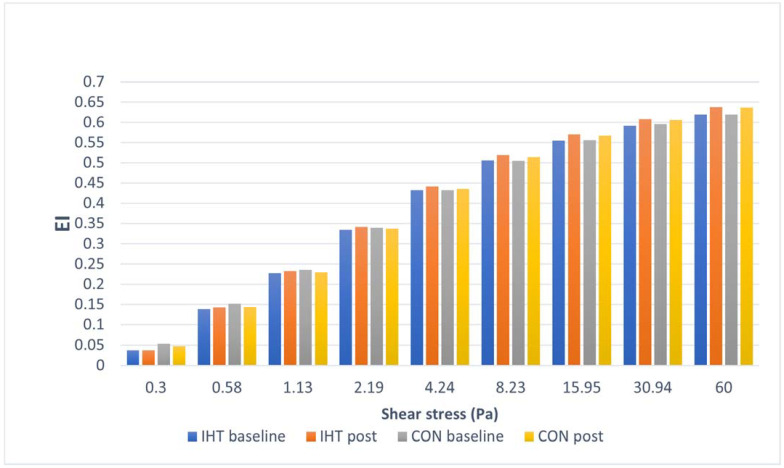
Red blood cell deformability (elongation index, EI) at particular shear stress values in rowers training in the hypoxia (intermittent hypoxic training, IHT) and normoxia (control, CON) conditions before and after the intervention.

**Figure 3 biology-11-01513-f003:**
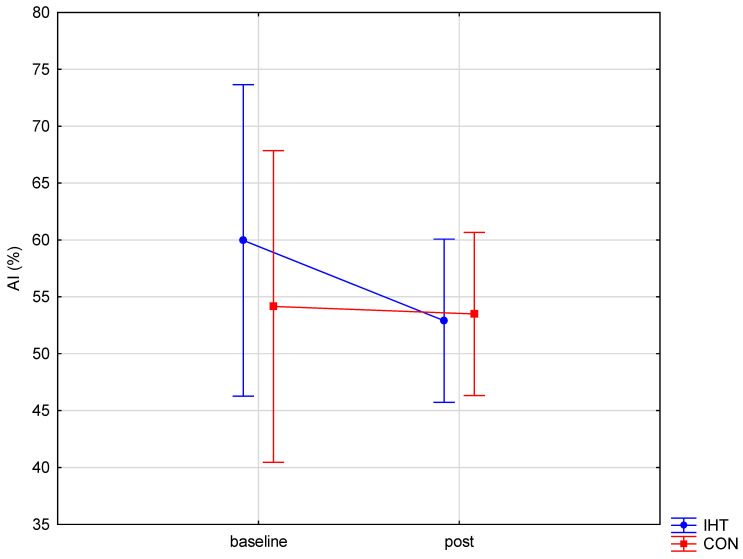
Red blood cell aggregation index (AI) in rowers training in the hypoxia (intermittent hypoxic training, IHT) and normoxia (control, CON) conditions before and after the intervention.

**Figure 4 biology-11-01513-f004:**
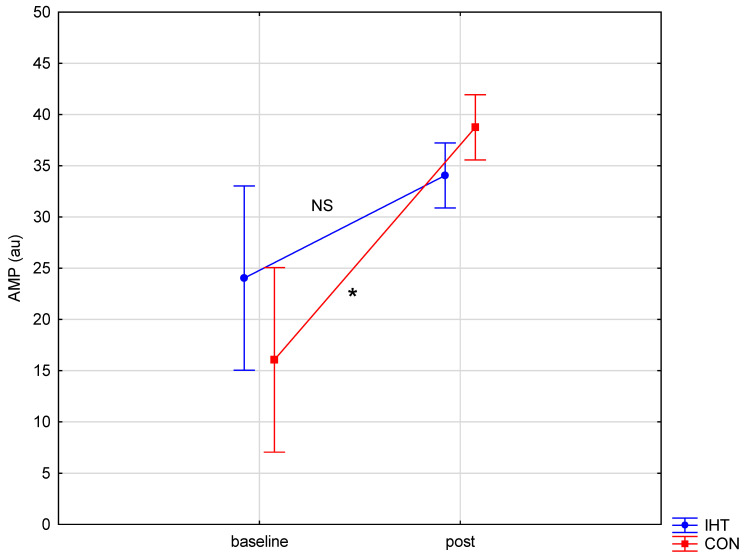
Red blood cell aggregation amplitude (AMP) in rowers training in the hypoxia (intermittent hypoxic training, IHT) and normoxia (control, CON) conditions before and after the intervention (*: *p* < 0.05, NS: non-significant).

**Figure 5 biology-11-01513-f005:**
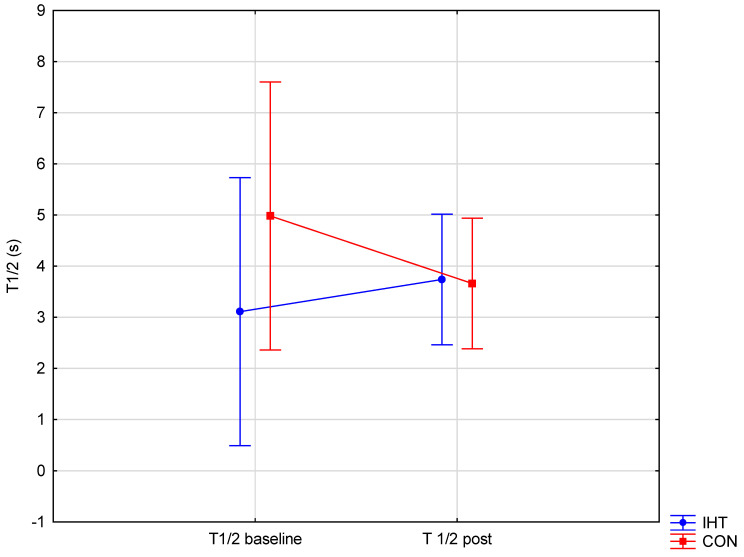
Half-time of total red blood cell aggregation (T1/2) in rowers training in the hypoxia (intermittent hypoxic training, IHT) and normoxia (control, CON) conditions before and after the intervention.

**Table 1 biology-11-01513-t001:** Selected somatic indicators of the examined competitors.

Group	Age [Years]	BH [cm]	BM [kg]	BMI [kg/m^2^]	F [%]
IHT	22.17 ± 1.83	185.5 ± 6.83	83.35 ± 9.04	24.18 ± 1.65	12.62 ± 2.96
CON	21.5 ± 1.64	186.83 ± 7.28	84.9 ± 12.14	24.22 ± 1.98	15.3 ± 2.42

BH—body height; BM—body mass; BMI—body mass index; F [%]—percentage of fat tissue; IHT—intermittent hypoxic training; CON—control group.

**Table 2 biology-11-01513-t002:** The 3-week training program.

Day of Week	Microcycle 1	Microcycle 2	Microcycle 3
1	Morning: rowing ergometer, continuous training, 50 min Afternoon: water training, continuous rowing, 90 min—recovery training (up to 75% HR_LT_)	Morning: rowing ergometer, continuous training, 50 min Afternoon: gym, strength endurance training, 90 min	Morning: rowing ergometer, continuous training, 50 min Afternoon: water training, continuous rowing, 90 min—recovery training (up to 75% HR_LT_)
2	Morning: off Afternoon: team games, basketball + football, 120 min	Morning: general development training, stabilization exercises, 60 min Afternoon: team games, basketball + football, 120 min	Morning: off Afternoon: team games, basketball + football, 120 min
3	Morning: rowing ergometer, 5 × 12 min Afternoon: gym, maximum strength training (10 exercises), 70 min	Morning: rowing ergometer, 5 × 12 min Afternoon: water training, speed training, 50 min	Morning: rowing ergometer, 5 × 12 min Afternoon: gym, maximum strength training (10 exercises), 70 min
4	Morning: general development training, stabilization exercises Afternoon: water training, interval training, 75 min	Morning: gym, maximum strength training (10 exercises), 70 min Afternoon: off	Morning: general development training, stabilization exercises Afternoon: water training, interval training, 75 min
5	Morning: rowing ergometer, continuous training, 50 min Afternoon: off	Morning: rowing ergometer, continuous training, 50 min Afternoon: water training, continuous rowing, 90 min—recovery training (up to 75% HR_LT_)	Morning: rowing ergometer, continuous training, 50 min Afternoon: off
6	Morning: cross-country running, 80 min (aerobic endurance training, 75–85% HR_LT_) Afternoon: strength endurance training, 90 min	Day off	Morning: cross-country running, 80 min (aerobic endurance training, 75–85% HR_LT_) Afternoon: strength endurance training, 90 min
7	Day off	Morning: cross-country running, 80 min (aerobic endurance training, 75–85% HR_LT_)	Day off

HR_LT_—threshold heart rate achieved at the lactate threshold in a graded test on a rowing ergometer.

**Table 3 biology-11-01513-t003:** Mean values (±SD) of blood morphology indicators in rowers training in the hypoxia and normoxia (control) conditions before and after the intervention.

Variable	Group	Baseline	Post	Effect: Group F (*p*)	Effect: Time F (*p*)	Interaction F (*p*)	Time Change: Pre vs. Post (*p*)
WBC [10^9^/L]	IHT	8.77 ± 2.56	6.37 ± 0.55	1.20 (0.3)	6.84 (0.02)	2.72 (0.13)	0.05 *
	CON	7.14 ± 0.79	6.59 ± 1.08				0.90 (NS)
	BD: post hoc (*p*)	0.25	0.99				
NEUT [10^9^/L]	IHT	5.06 ± 2.06	3.42 ± 0.28	2.34 (0.15)	1.69 (0.22)	5.18 (0.04) *	0.11 (NS)
	CON	3.27 ± 0.40	3.72 ± 0.94				0.89 (NS)
	BD: post hoc (*p*)	0.05 *	0.96				
LYM [10^9^/L]	IHT	3.25 ± 1.57	2.09 ± 0.36	0.42 (0.53)	11.25 (0.007) *	0.25 (0.62)	0.08 (NS)
	CON	2.85 ± 0.41	1.99 ± 0.19				0.24 (NS)
	BD: post hoc (*p*)	0.84	0.99				
MONO [10^9^/L]	IHT	0.72 ± 0.19	0.64 ± 0.13	0.06 (0.80)	3.02 (0.11)	0.43 (0.52)	–
	CON	0.68 ± 0.08	0.64 ± 0.18				–
	BD: post hoc (*p*)	–	–				
EOS [10^9^/L]	IHT	0.15 ± 0.08	0.17 ± 0.07	2.20 (0.16)	1.75 (0.21)	6.78 (0.02) *	0.17 (NS)
	CON	0.29 ± 0.16	0.21 ± 0.12				0.07 (NS)
	BD: post hoc (*p*)	0.17	0.92				
BASO [10^9^/L]	IHT	0.05 ± 0.02	0.06 ± 0.02	0.31 (0.58)	0.26 (0.61)	1.45 (0.25)	–
	CON	0.05 ± 0.02	0.045 ± 0.02				–
	BD: post hoc (*p*)	–	–				
RBC [10^12^/L]	IHT	5.15 ± 0.2	5.14 ± 0.28	0.003 (0.95)	3.08 (0.10)	0.06 (0.80)	–
	CON	5.05 ± 0.2	5.07 ± 0.34				–
	BD: post hoc (*p*)	–	–				
HGB [g/L]	IHT	14.85 ± 0.59	14.73 ± 0.76	0.60 (0.45)	0.90 (0.36)	0.03 (0.87)	–
	CON	15.22 ± 0.82	15.05 ± 0.98				–
	BD: post hoc (*p*)	–	–				
HCT [L/L]	IHT	43.23 ± 1.72	42.96 ± 2.33	0.04 (0.83)	1.36 (0.27)	0.27 (0.61)	–
	CON	43.73 ± 2.67	43.03 ± 2.85				–
	BD: post hoc (*p*)	–	–				
MCV [fL]	IHT	84.02 ± 3.98	85.12 ± 4.34	0.04 (0.84)	2.06 (0.18)	3.16 (0.10)	–
	CON	85.02 ± 2.87	84.9 ± 2.5				–
	BD: post hoc (*p*)	–	–				
MCH [fmol]	IHT	28.87 ± 1.25	29.58 ± 0.75	1.40 (0.26)	1.90 (0.19)	0.68 (0.42)	–
	CON	29.2 ± 0.83	29.67 ± 0.67				–
	BD: post hoc (*p*)	–	–				
MCHC [mmol/L]	IHT	34.35 ± 0.27	34.35 ± 1.06	2.11 (0.17)	0.14 (0.71)	0.14 (0.71)	–
	CON	34.82 ± 0.47	35.0 ± 1.04				–
	BD: post hoc (*p*)	–	–				
RDW-SD [fL]	IHT	38.68 ± 2.56	40.55 ± 3.0	3.31 (0.09)	1.78 (0.21)	1.04 (0.33)	–
	CON	37.32 ± 2.38	37.57 ± 1.83				–
	BD: post hoc (*p*)	–	–				
PLT [10^9^/L]	IHT	245.67 ± 45.48	240 ± 21.94	2.22 (0.16)	0.21 (0.65)	1.61 (0.23)	–
	CON	271.17 ± 38.49	283.33 ± 54.2				–
	BD: post hoc (*p*)	–	–				
MPV [fL]	IHT	10.63 ± 0.79	10.5 ± 0.84	0.002 (0.96)	0.98 (0.34)	0.02 (0.87)	–
	CON	10.63 ± 0.84	10.45 ± 1.16				–
	BD: post hoc (*p*)	–	–				
PCT [%]	IHT	0.25 ± 0.05	0.25 ± 0.05	3.76 (0.08)	1.0 (0.34)	1.0 (0.34)	–
	CON	0.32 ± 0.04	0.3 ± 0.06				–
	BD: post hoc (*p*)	–	–				
PDW [fL]	IHT	13.02 ± 2.12	12.52 ± 1.52	0.03 (0.85)	1.47 (0.25)	0.13 (0.71)	–
	CON	12.68 ± 1.73	12.42 ± 2.74				–
	BD: post hoc (*p*)	–	–				

*: Significantly different when compared with controls (*p* < 0.05). WBC—white blood cells; NEUT—neutrophils; LYM—lymphocytes; MONO—monocytes; EOS—eosinocytes; BASO—basophils; RBC—red blood cells; HGB—haemoglobin; HCT—haematocrit; MCV—mean corpuscular volume; MCH—mean corpuscular haemoglobin; MCHC—mean corpuscular haemoglobin concentration; RDW-SD—red blood cell distribution width standard deviation; PLT—platelets; MPV—mean platelet volume; PCT—procalcitonin; PDW—platelet distribution width; BD—between-group difference; NS–nonsignificant; IHT—intermittent hypoxic training; CON—control group.

**Table 4 biology-11-01513-t004:** Mean values (±SD) of the elongation index in rowers training in the hypoxia and normoxia (control) conditions before and after the intervention.

Variable	Group	Baseline	Post	Effect: Group F (*p*)	Effect: Time F (*p*)	Interaction F (*p*)	Time Change: Pre vs. Post (*p*)
EI at 0.30 Pa	IHT	0.037 ± 0.01	0.037 ± 0.01	3.17 (0.10)	0.70 (0.42)	0.53 (0.48)	–
	CON	0.053 ± 0.01	0.047 ± 0.01				–
	BD: post hoc (*p*)	–	–				
EI at 0.58 Pa	IHT	0.138 ± 0.01	0.142 ± 0.02	0.608 (0.45)	0.709 (0.42)	3.08 (0.10)	–
	CON	0.152 ± 0.01	0.143 ± 0.02				–
	BD: post hoc (*p*)	–	–				
EI at 1.13 Pa	IHT	0.227 ± 0.01	0.232 ± 0.01	0.102 (0.75)	0.06 (0.80)	1.98 (0.18)	–
	CON	0.235 ± 0.01	0.229 ± 0.02				–
	BD: post hoc (*p*)	–	–				
EI at 2.19 Pa	IHT	0.334 ± 0.01	0.341 ± 0.01	0.07 (0.93)	0.98 (0.36)	2.51 (0.14)	–
	CON	0.339 ± 0.01	0.337 ± 0.02				–
	BD: post hoc (*p*)	–	–				
EI at 4.24 Pa	IHT	0.432 ± 0.01	0.441 ± 0.01	0.41 (0.53)	5.45 (0.04) *	1.40 (0.26)	0.12 (NS)
	CON	0.432 ± 0.01	0.435 ± 0.01				0.84 (NS)
	BD: post hoc (*p*)	0.99	0.70				
EI at 8.23 Pa	IHT	0.506 ± 0.01	0.519 ± 0.01	0.43 (0.52)	31.73 (0.0002) *	1.31 (0.27)	0.003 *
	CON	0.505 ± 0.01	0.514 ± 0.01				0.04 *
	BD: post hoc (*p*)	0.99	0.72				
EI at 15.95 Pa	IHT	0.555 ± 0.01	0.570 ± 0.01	0.05 (0.82)	65.55 (0.00001) *	1.0 (0.34)	0.0005 *
	CON	0.556 ± 0.01	0.567 ± 0.01				0.002 *
	BD: post hoc (*p*)	0.99	0.94				
EI at 30.94 Pa	IHT	0.592 ± 0.01	0.608 ± 0.00	0.0 (0.88)	131.4 (0.00) *	5.4 (0.04) *	0.0002 *
	CON	0.596 ± 0.00	0.606 ± 0.00				0.0005 *
	BD: post hoc (*p*)	0.88	0.93				
EI at 60.00 Pa	IHT	0.619 ± 0.00	0.638 ± 0.00	0.1 (0.75)	197.4 (0.00) *	0.6 (0.47)	0.0002 *
	CON	0.619 ± 0.00	0.637 ± 0.00				0.0002 *
	BD: post hoc (*p*)	0.27	0.69				

*: Significantly different when compared with controls (*p* < 0.05). EI—elongation index; BD—between-group difference; NS—nonsignificant; IHT—intermittent hypoxic training; CON—control group.

**Table 5 biology-11-01513-t005:** Mean values (±SD) of the aggregation index, half-time of total aggregation, and erythrocyte aggregation amplitude in rowers training in the hypoxia and normoxia (control) conditions before and after the intervention.

Variable	Group	Baseline	Post	Effect: Group F (*p*)	Effect: Time F (*p*)	Interaction F (*p*)	Time Change: Pre vs. Post (*p*)
AI [%]	IHT	59.96 ± 20.36	52.89 ± 7.39	0.34 (0.57)	0.52 (0.48)	0.36 (0.55)	–
	CON	54.15 ± 6.23	53.49 ± 8.36				–
	BD: post hoc (*p*)	–	–				
T1/2 [s]	IHT	3.11 ± 1.94	3.72 ± 1.16	1.27 (0.28)	0.11 (0.74)	0.88 (0.37)	–
	CON	4.98 ± 3.58	3.66 ± 1.61				–
	BD: post hoc (*p*)	–	–				
AMP [au]	IHT	24.03 ± 12.22	34.05 ± 3.57	0.26 (0.61)	32.16 (0.0002) *	4.82 (0.05) *	0.12 (NS)
	CON	16.05 ± 6.82	38.74 ± 3.42				0.001 *
	BD: post hoc (*p*)	0.27	0.69				

*: Significantly different when compared with controls (*p* < 0.05). AI—aggregation index; T1/2—half-time of total aggregation; AMP—erythrocyte aggregation amplitude; BD—between-group difference; NS—nonsignificant; IHT—intermittent hypoxic training; CON—control group.

**Table 6 biology-11-01513-t006:** Mean values (±SD) of fibrinogen concentrations in rowers training in the hypoxia and normoxia (control) conditions before and after the intervention.

Variable	Group	Baseline	Post	Effect: Group F (*p*)	Effect: Time F (*p*)	Interaction F (*p*)	Time Change: Pre vs. Post (*p*)
Fibrinogen [g/L]	IHT	1.95 ± 0.47	2.16 ± 0.24	2.37 (0.15)	2.23 (0.16)	0.009 (0.92)	–
	CON	2.06 ± 0.17	2.30 ± 0.22				–
	BD: post hoc (*p*)	–	–				

BD—between-group difference; IHT—intermittent hypoxic training; CON—control group.

**Table 7 biology-11-01513-t007:** Mean values (±SD) of acetylcholinesterase activity in rowers training in the hypoxia and normoxia (control) conditions before and after the intervention.

Variable	Group	Baseline	Post	Effect: Group F (*p*)	Effect: Time F (*p*)	Interaction F (*p*)	Time Change: Pre vs. Post (*p*)
AChE [IU/g HGB]	IHT	32.03 ± 5.62	36.35 ± 4.04	0.26 (0.61)	6.98 (0.024) *	1.09 (0.32)	0.10 (NS)
	CON	35.17 ± 1.63	37.03 ± 8.69				0.68 (NS)
	BD: post hoc (*p*)	0.85	0.99				

* Significantly different when compared with controls (*p* < 0.05). AChE—acetylcholinesterase; HGB—haemoglobin; BD—between-group difference; NS—nonsignificant; IHT—intermittent hypoxic training; CON—control group.

## Data Availability

The datasets used and/or analysed during the current study are available from the corresponding author on reasonable request.

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
