# Peer review of "Effect of Intermittent Hypoxic Training on Selected Biochemical Indicators, Blood Rheological Properties, and Metabolic Activity of Erythrocytes in Rowers"

_biology, 2022, doi:10.3390/biology11101513_

Round 1

Reviewer 1 Report

Dear Authors,

Thank you for your efforts in your research. Your research is valuable in terms of its subject, scope and content. I would also like to thank you for your contributions to the field. Your research does not require a major or minor correction. There are only some grammatical errors in English, once the grammatical errors are corrected, your research is eligible for publication in the journal Biology. congratulations

Best

Author Response

Dear Reviewer

Thank You for your positive review.

Best regards

Autors

Reviewer 2 Report

This study evaluated the effect of a 3-week intermittent hypoxic training on blood biochemical indicators, blood rheological properties, erythrocyte enzymatic activity, and maximal oxygen uptake in advanced and elite competitive rowers. There is some novelty here; however, the manuscript has deep flaws in the writing, presentation of paragraphs, and results. Lastly, the conclusion is not supported by the results, at least in the way that was presented. Several sentences lack citations, and explanations for several methods are required. My comments are intended to help the authors improve the study. If properly adjusted, it can be revised again.

Introduction

-Line 39-40 – This is vague and non-scientific.

-Line 41 – Crew type does not sound well. Consider changing rowing disciplines.

-Line 42 – “..based mainly..” can be misleading. Change to significant, large, or similar words.

-Line 43 – “must demonstrate” does not sound well.

-Line 44 – There are fundamental mistakes in this sentence. The MLSS is the gold standard protocol for measuring aerobic capacity, while anaerobic power (VO2max) has a distinct perspective, although associated with MLSS.

-Line 46 – “indicators” does not sound well. Also, some citation is missing here.

-Based on these initial comments, I recommend authors search for specialized proofreading services. Although a few grammar mistakes were noticed, several words must be replaced.

-Line 46-47 – Missing citation.

-Line 47-48 – Missing citation.

-Line 50 -Missing citation.

-Based on the three previous comments, I recommend a deep revision regarding citations in the manuscript. Moreover, other sentences are misleading considering the citation (Line 79-81 – Citation 18).

Material and methods

-Include more information on the rowers (e.g. weekly training load frequency, participation in national or international competitions, international ranking, etc..). This is very important, once you classified them as advanced and elite.

-Explain the criteria for inclusion.

-Explain the criteria for exclusion. What do recent injuries mean? The same for “training breaks.

-Line 101 – At this point readers have no information about the 2-km rowing test. Here you need to provide a well-elaborated figure regarding your experimental design. For instance, in lines 102-104 you mention the hypobaric condition. How this was performed. The authors need to reorganize this entire section.

-Table 1 – What do decimal cases in age mean?

-Lines 127-130 – Authors need to present the intensity and volume of each training session and its components.

-Line 131- provide the citation for this incremental protocol.

-Line 134 – Explain the criteria for lactate threshold determination. There are other methods for this purpose with higher precision.

-Line 136 – This entire paragraph is confusing. The figure that I have suggested is also welcome here.

-Line 152 – Provide the location where blood samples were taken, as well as the volume.

-Line 171 – There is no reason to provide a separate paragraph for this information. Short paragraphs reduce the elegance of your manuscript.

-Line 181- Which ANOVA was used? One-way or Two-way? Provide the dependent and independent variables.

Results, discussion, and conclusion

-Line 188 -You have just mentioned the statistical test. There is no reason to repeat this here.

-The quality of every figure can be improved. There is information missing in the legends.

-Tables – It is not clear what “Time Change” means.

-Authors conclude that “The reported findings demonstrate that the phenomenon of hypoxia can be an extremely effective training measure in rowers as IHT exerted a beneficial impact on the 386 athletes’ performance”. Several p-values for the interaction are close to 5%. Here you have great possibilities of assuming something that is not true. Actually, I disagree with your conclusion, at least the intensity that you mentioned it.

-Be fair to your results. The “Time Change” column is confusing and does not sustain your conclusions. The entire results section is confusing, wordy, and poorly presented. The entire section must be adjusted, providing new figures and the effect sizes of comparisons.

-The discussion requires deep improvements. The first paragraph does not summarize the findings, and the aim is brought in the second. Few relevant discussions were provided here.

Author Response

Dear Reviewer

Thank you for your suggestions.

Authors

Reviewer 3 Report

Please find the attchment.

Author Response

(The authors gave the same response as above.)

Round 2

Reviewer 2 Report

The authors answered all my questions, in addition to including more references throughout the study.

 On the other hand, they did not add the figures I recommended, nor did they adjust the acronyms in the legends. The response given was:

 "Thank you for your comment. In our opinion, the figures are readable and all abbreviations are explained in the captions".

Just because it's "readable" doesn't mean its quality is high. Anyway, my suggestion was made to amplify the impact of the article, contributing both to the post-publication moment for the authors and for Biology. Thus, the choice is up to the authors themselves. The figures are still of poor quality. Additionally, some acronyms have not yet been included in the captions. Figures need to be self-explanatory. However, I reiterate that this care must be taken by the authors. The option is theirs.

Finally, my question about the decimal places in the age variable is rhetorical. The authors' response is not enough, but, again, it is their names that are in evidence in the article.

Anyway, I believe that within its limitations, the study contributes to the scope of exercise physiology.

Author Response

Dear Reviewer

I send response revision.

Author

Reviewer 3 Report

My all concerns were addressed, excepted by one. The justification provided by authors regarding the absence of sample size must be included in the study's methods.

Author Response

Dear Reviewer

Thank you again for your helpful comments. We have added sample size justification to the methods.

Author
